# Uncovering Strong Lottery Tickets in Graph Transformers: A Path to Memory Efficient and Robust Graph Learning

**Hiroaki Ito**[1]                                        *ito.hiroaki@artic.iir.isct.ac.jp*

**Jiale Yan**[1]                                          *yan.jiale@artic.iir.isct.ac.jp*

**Hikari Otsuka**[1]                                      *otsuka.hikari@artic.iir.isct.ac.jp*

**Kazushi Kawamura**[1]                                   *kawamura@artic.iir.isct.ac.jp*

**Masato Motomura**[1]                                    *motomura@artic.iir.isct.ac.jp*

**Thiem Van Chu**[1]                                      *thiem@ict.eng.isct.ac.jp*

**Daichi Fujiki**[1]                                      *dfujiki@artic.iir.isct.ac.jp*

[1] *Department of Information and Communications Engineering, Institute of Science Tokyo, Japan*

**Reviewed on OpenReview:** *https://openreview.net/forum?id=B1q9po4LPl*

## Abstract

Graph Transformers (GTs) have recently demonstrated strong capabilities for capturing complex relationships in graph-structured data using global self-attention mechanisms. On the other hand, their high memory requirements during inference remain a challenge for practical deployment. In this study, we investigate the existence of strong lottery tickets (SLTs) — subnetworks within randomly initialized neural networks that can attain competitive accuracy without weight training — in GTs. Previous studies have explored SLTs in message-passing neural networks (MPNNs), showing that SLTs not only exist in MPNNs but also help mitigate over-smoothing problems and improve robustness. However, the potential of SLTs in GTs remains unexplored. With GTs having $4.5\times$ more parameters than MPNNs, SLTs hold even greater application value in this context. We find that fixed random weights with a traditional SLT search method cannot adapt to imbalances of features in GTs, leading to highly biased attention that destabilizes model performance. To overcome this issue and efficiently search for SLTs, we introduce a novel approach called Adaptive Scaling. We empirically confirm the existence of SLTs within GTs and demonstrate their versatility through extensive experiments across different GT architectures, including NodeFormer, GRIT, and GraphGPS. Our findings demonstrate that SLTs achieve comparable accuracy while reducing memory usage by $2$–$32\times$, effectively generalize to out-of-distribution data, and enhance robustness against adversarial perturbations. This work highlights that SLTs offer a resource-efficient approach to improving the scalability, efficiency, and robustness of GTs, with broad implications for applications involving graph data.

## 1 Introduction

Graph Neural Networks (GNNs) have achieved remarkable success in modeling graph-structured data across diverse domains such as social networks, biological networks, and recommendation systems (Fan et al., 2019; Choi et al., 2021; Wu et al., 2022b). By leveraging node connectivity and feature aggregation, GNNs excel at capturing local dependencies within graphs, enabling them to solve complex real-world tasks.

However, traditional GNNs—*a.k.a.* message passing neural networks (MPNNs)—face significant limitations when modeling long-range dependencies. As MPNNs rely on iterative message passing between neighboring nodes, capturing interactions between distant nodes often requires stacking many layers, which leads to challenges such as over-smoothing (Li et al., 2018; Oono & Suzuki, 2020) and over-squashing (Topping et al., 2022; Di Giovanni et al., 2023).

Graph Transformers (GTs) have emerged as a powerful alternative to overcome the above problems, incorporating global self-attention mechanisms and achieving SOTA accuracy. Specifically, it allows nodes to attend to all other nodes in the graph. This global perspective enables GTs to effectively capture long-range dependencies, making them well-suited for tasks involving distant interactions across large and complex graphs.

Despite their high performance, GTs have the drawback of increased memory demands due to their multiple linear transformations during self-attention operations. For example, NodeFormer (Wu et al., 2022a) requires 4.59 times more memory than GCN (Kipf & Welling, 2017), and the linear layers at the GT layers take 89 % of the model memory size. This resource intensity poses practical challenges for deploying these models in environments with limited hardware capacity. Moreover, when deploying GTs in real-world applications, robustness to perturbations (Zhu et al., 2024; Foth et al., 2024) and generalization to out-of-distribution (OOD) data (Li et al., 2022; Wu et al., 2023) are critical concerns. Models must be reliable in the face of noisy data and unexpected inputs.

To tackle these issues, network pruning techniques have been explored to reduce the model size while keeping competitive accuracy. Among these, the Strong Lottery Ticket Hypothesis (SLTH) (Zhou et al., 2019; Ramanujan et al., 2020; Malach et al., 2020) posits that within a randomly initialized neural network, there exist subnetworks—referred to as Strong Lottery Tickets (SLTs)—that can achieve competitive accuracy without any weight training. Recent work by Huang et al. (2022) demonstrated the existence of SLTs in MPNNs. It has also been observed that SLTs can mitigate over-smoothing, a common issue leading to accuracy loss in MPNNs, and enhance the robustness and OOD performance. However, it remains an open question whether accurate SLTs exist within GTs and, if so, whether they offer benefits in perturbation and OOD challenges.

In this work, we address these questions by investigating the existence of SLTs within GTs and their utility. During the investigation, we found that typical fixed random weights with a traditional SLT search method cannot adapt to distribution imbalances of activations, leading to highly skewed attention destabilizing model performance. We introduce a novel approach incorporating Adaptive Scaling to overcome the challenges and effectively search for SLTs. Moreover, our proposed SLT methods reduce the memory footprint of GT and enhance their robustness and generalization capabilities.

We conduct comprehensive experiments across two main kinds of GT architectures to validate our proposed approach: hybrid GTs (*e.g.*, GraphGPS (Rampášek et al., 2022)), which integrate MPNNs into GTs, and pure GTs (*e.g.*, NodeFormer (Wu et al., 2022a) and GRIT (Ma et al., 2023)), which do not have MPNNs. Our results empirically confirm the presence of SLTs in GTs and demonstrate that these subnetworks achieve comparable or even superior performance to trained dense models while significantly reducing memory usage. Furthermore, we show that SLTs enhance robustness against adversarial perturbations and improve generalization to OOD data. This finding is particularly significant for real-world applications where data is often noisy or deviates from the training distribution, and model reliability is important.

Our key contributions are summarized as follows:

- For the first time, we reveal the existence of SLTs in GTs by utilizing Adaptive Scaling methods, which combine Dynamic Weight Scaling (DWS) and Attention Stability Norm (ASNorm) to balance the attention scores and enhance training stability. Unlike the existing SLT search method, which fails to identify lottery tickets, our approach successfully uncovered them with $\geq 80\%$ sparsity.

- Our analysis reveals that the identified SLTs in GTs contribute to increased robustness against adversarial perturbations and improved generalization to OOD data. This highlights the potential of SLTs for building more reliable and safe GT models for practical applications.

- Our experiments cover hybrid (with MPNN) and pure (without MPNN) GTs, evaluated on seven mainstream datasets for node-level tasks and six for graph-level tasks. Compared to fully trained dense models, our approach achieves a 2–32× reduction in memory consumption while maintaining comparable accuracy and providing higher robustness.

## 2 Related Work

**Graph Neural Networks:**   Graph Neural Networks (GNNs) are designed to learn optimal representations of graph-structured data, where nodes represent entities and edges capture the relationships between them. GNNs have been widely applied across various real-world domains, including social network analysis (Fan et al., 2019), recommendation systems (Wu et al., 2022b; Gao et al., 2022), drug discovery (Gaudelet et al., 2021), and autonomous driving(Choi et al., 2021). Graph Transformers (GTs) have emerged as an advanced approach to GNNs. By leveraging the self-attention mechanism of Transformers, they can capture local and global dependencies within a graph. Unlike MPNNs, which focus on aggregating information from local neighborhoods, GTs can attend to all nodes in the graph. This capability allows them to address the limitations of MPNNs like over-smoothing (Li et al., 2018; Oono & Suzuki, 2020) where node representations become indistinguishable, and over-squashing (Topping et al., 2022; Di Giovanni et al., 2023) where information from distant nodes is inadequately captured. They also perform better in tasks that require a global understanding of the graph structure, such as large graphs (Lerer et al., 2019; Xue et al., 2023) or time series data (Jiang & Luo, 2022; Zhang et al., 2022).

On the other hand, GTs face several significant challenges, particularly in terms of generalization (Li et al., 2022; Wu et al., 2023), robustness (Zhu et al., 2024; Foth et al., 2024), and memory demands (Ku, 2023). They often struggle to generalize to unseen graphs, especially when they have different distributions of features or classes during inference. Additionally, GTs are susceptible to adversarial attacks and noisy data, decreasing accuracy and unreliable results. Furthermore, their complex self-attention layers result in high memory demands, limiting scalability in practical applications. Addressing these issues is essential for improving the applicability of GTs in real-world tasks.

**Strong Lottery Ticket Hypothesis:**   The Lottery Ticket Hypothesis (LTH) (Frankle & Carbin, 2019) proposes that within a dense neural network, there exist sparse subnetworks that, when trained in isolation, can achieve comparable or even higher accuracy than the original network. Building on this, Zhou et al. (2019) identified an even stronger phenomenon, the Strong Lottery Ticket Hypothesis (SLTH), which conjectures that accurate subnetworks exist within a randomly initialized network and can achieve comparable performance without any weight updates. However, it was difficult to identify such subnetworks due to the vast combinatorial search space, resulting in limited performance. To address this challenge, Ramanujan et al. (2020) introduced the Edge-Popup method, efficiently searching for accurate subnetworks within randomly initialized networks without weight training. Subsequent studies have confirmed the existence of SLT experimentally (Zhou et al., 2021b; Sreenivasan et al., 2022; Otsuka et al., 2025) and theoretically (Malach et al., 2020; Orseau et al., 2020; Pensia et al., 2020).

Some works, such as Pensia et al. (2020) and Orseau et al. (2020), provide theoretical guarantees for SLT in over-parameterized neural networks. Their proofs typically rely on constructing sparse subnetworks in simplified architectures (e.g., two-layer ReLU networks) to show comparable performance to the full models. Extending these theoretical insights to Transformers remains non-trivial due to architectural differences, particularly the multi-head self-attention mechanism.

In general Transformers, initial empirical research has been conducted on LTH (Brix et al., 2020) and SLTH (Shen et al., 2021). However, the theoretical analysis of SLTs in Transformers remains unresolved. General Transformers and GTs share architectural components like attention mechanisms and feed-forward networks. Still, the differences in their input data, sequential text versus graph structures, introduce unique challenges for applying the concept of SLTs. For instance, GTs must handle irregular connectivity and sparsity, which differ significantly from the structured redundancies in NLP tasks. While the Edge-Popup could theoretically be applied to GTs, its naive application fails due to these challenges.

In the domain of GNNs, Huang et al. (2022) was the first to extend the SLTH to MPNNs. They explored the existence of SLTs within MPNNs and demonstrated their potential to address the over-smoothing problem (Li et al., 2018; Oono & Suzuki, 2020) and improve robustness (Zhu et al., 2024; Foth et al., 2024) and out-of-distribution (OOD) performance (Li et al., 2022; Wu et al., 2023). Additionally, Yan et al. (2024) advanced the application of SLTH in MPNNs by multicoating (Okoshi et al., 2022) and folding (García-Arias et al., 2023) supermasks, significantly reducing memory requirements while maintaining competitive accuracy. Collectively, these studies demonstrate the potential of SLTs in the context of MPNNs. However, the existence and utility of SLTs within GTs remain unexplored.

Therefore, we aim to investigate whether SLTs can alleviate the memory limitations of GTs and enhance their robustness and generalization capabilities. By proposing new modules tailored for graph-structured data, such as ASNorm and DWS, we empirically demonstrate the existence of SLTs in GTs and highlight the need for future theoretical advancements in this domain.

# 3 Strong Lottery Tickets within Graph Transformers

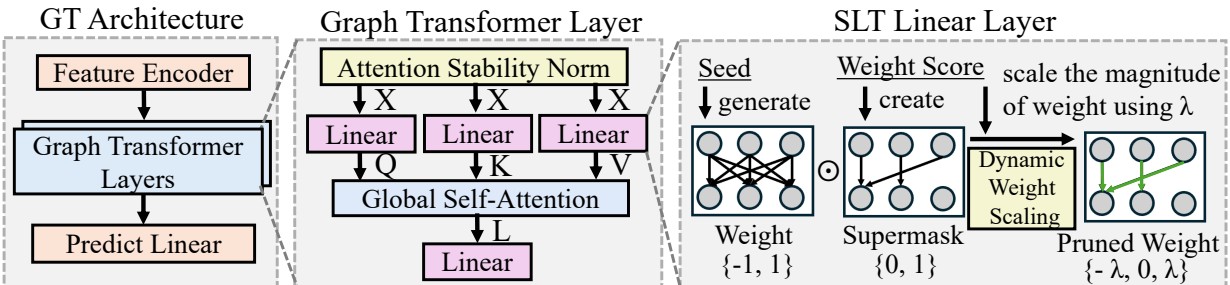

Figure 1: Overview of the Edge-Popup process using proposed Adaptive Scaling for GTs.

## 3.1 Preliminaries

**Notation:** In this paper, we adopt the following notational conventions: matrices are denoted by bold uppercase letters, $\mathbf{X}$, vectors by bold lowercase letters, $\mathbf{v}$, and scalars are written in regular lowercase letters, $s$. For the $(i, j)$-th element of a matrix $\mathbf{X}$, we use $\mathbf{X}_{ij}$. The graph's adjacency matrix is denoted as $\mathbf{A}$, while $\mathbf{X}$ represents the node feature matrix. Additionally, $\mathbf{W}$ refers to the weight matrix.

**Graph Transformers (GTs):** The architecture of GTs can be divided into two categories: pure GTs, which do not include MPNNs, and hybrid GTs, which do include MPNNs. This work uses NodeFormer and GRIT as pure GTs and GraphGPS as a hybrid GT. NodeFormer utilizes a kernelized Gumbel-Softmax operator to enable efficient all-pair message passing, reducing the computational complexity from quadratic to linear. GRIT incorporates graph inductive biases without using message passing using learned relative positional encodings initialized with random walk probabilities. GraphGPS leverages positional and structural encoding for efficient node representation and uses both the MPNN and GT layers simultaneously.

GTs calculate the attention score of each node in a graph by global self-attention mechanism:

$$\text{Attn}(\mathbf{Q}, \mathbf{K}, \mathbf{V}) = \text{Softmax}\left(\frac{\mathbf{Q}\mathbf{K}^{\mathrm{T}}}{\sqrt{d_k}}\right)\mathbf{V}, \tag{1}$$

$$\text{Softmax}(z_i) = \frac{\exp(z_i)}{\sum_{j=1}^{N}\exp(z_j)}, \tag{2}$$

$$\mathbf{Q} = \mathbf{H}\mathbf{W}_Q, \quad \mathbf{K} = \mathbf{H}\mathbf{W}_K, \quad \mathbf{V} = \mathbf{H}\mathbf{W}_V, \tag{3}$$

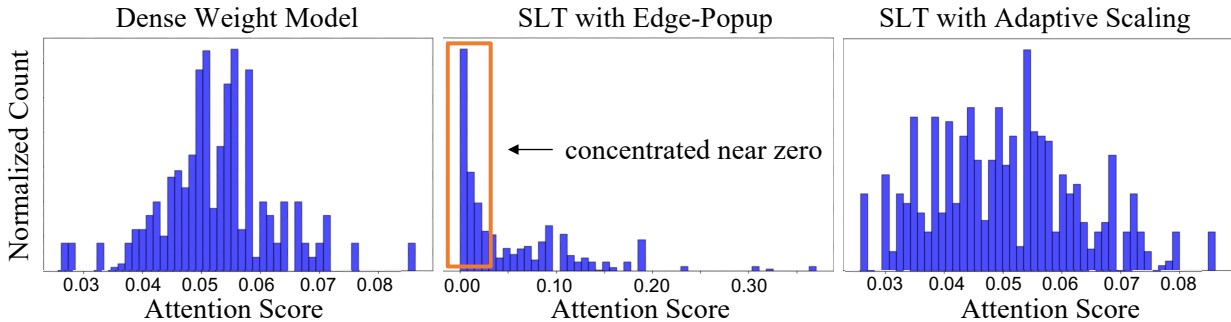

Figure 2: Comparing the attention score distributions between dense models and SLTs with and without Adaptive Scaling.

where $\mathbf{H} \in \mathbb{R}^{N \times d_h}$ represents the hidden representation of the nodes, and $N$ is the number of nodes. $\mathbf{W}_Q, \mathbf{W}_K, \mathbf{W}_V \in \mathbb{R}^{d_h \times d_k}$ project the hidden representation into the query, key, and value spaces, respectively. After global self-attention, the immediate results are further transformed through a linear projection with $\mathbf{W}_L$.

**Edge-Popup:** We follow the Edge-Popup (Ramanujan et al., 2020; Sreenivasan et al., 2022) to search for SLTs in linear layers. Each linear layer is assigned learnable scores with the same shape as the weights. These scores indicate the importance of the network connections and are used to generate the supermask by selecting the top-$k\%$ with the highest scores. Here, $k \in [0, 100]$ is a hyperparameter that controls the density of the supermask. Each element of the supermask $m_{ij}$ is defined as:

$$m_{ij} = h(s_{ij}), \quad h(s_{ij}) = \begin{cases} 1, & |s_{ij}| \geq t \\ 0, & |s_{ij}| < t \end{cases}, \tag{4}$$

where $s_{ij}$ is the element of weight score matrix, and $t \in \mathbb{R}^+$ is the threshold corresponding to the top $k\%$ of the absolute values in the weight scores. This implies that the SLT has a sparsity level of $(100 - k)\%$.

The weights in the linear layers $(\mathbf{W}_Q, \mathbf{W}_K, \mathbf{W}_V, \mathbf{W}_L)$ are converted into an SLT format, denoted as $\mathbf{W}_{\text{SLT}}$, and are obtained by:

$$\mathbf{W}_{\text{SLT}} = \mathbf{W}_{\text{rand}} \odot \mathbf{M}, \tag{5}$$

where $\mathbf{W}_{\text{rand}} \in \{-1, 1\}^{m \times n}$ is a randomly initialized weight, and $\mathbf{M} \in \{0, 1\}^{m \times n}$ is the supermask for pruning.

### 3.2 Adaptive Scaling Strengthens SLTs

In this section, we show our approach to finding SLTs utilizing the Edge-Popup with our introduced Adaptive Scaling and evaluate its performance on accuracy to show its effectiveness.

**Edge-Popup Fails in Finding SLTs within GTs:** We observe that directly using Edge-Popup often fails to find accurate SLTs, leading to worse accuracy in some datasets. Upon examining the distribution of attention scores, we observe that some node activations in the Softmax calculation become disproportionately large. As shown in Equation (2), when certain values of the Softmax input $z_i$ are disproportionately large, the exponential term in the denominator causes other values to become extremely small, diminishing their contributions. As we explain below, a naive application of Edge-Popup often introduces such an imbalanced distribution of attention scores in GTs. As a result, these scores fail to provide meaningful attention to other nodes, leading to a distribution that deviates from an ideal form. This issue is particularly problematic in

deep-layer models, where the distribution imbalance accumulates layer by layer, destabilizing the scaling of activations. For example, when training GraphGPS on the CLUSTER dataset, we observed that in a 4-layer model, the accuracy difference between the dense model and SLT using Edge-Popup is 1%, 69.89% for the dense model and 68.49% for Edge-Popup. However, in a deeper 16-layer model, which is the recommended setting in their paper, the performance of Edge-Popup significantly deteriorates. The dense model achieves an accuracy of 78.02%, whereas the Edge-Popup model's accuracy drops drastically to 17.57%.

**Unbalanced Distribution of Attention Scores:** Figure 2 illustrates a comparison of attention score distributions between dense models and SLT using Edge-Popup. The attention scores are distributed like a normal distribution in the dense model. In contrast, the SLT with Edge-Popup exhibits a distribution where most attention values are concentrated near zero, while a few values are significantly large. This indicates that the model assigns extreme attention to some nodes, disrupting the balanced aggregation across the graph.

We explore why some attention scores get excessively large. Dense models can dynamically adapt their weights through gradient descent to maintain balanced activations. In contrast, SLT weights obtained via Edge-Popup are randomly initialized and remain fixed during training, preventing the model from adjusting weight magnitudes to counteract imbalances in activations. The fixed nature of SLT weights leads to disproportionate activations in the attention mechanism, causing certain nodes to dominate the attention distribution. This imbalance accumulates across layers, undermining the representational capacity of the model.

To mitigate this imbalance, we propose Adaptive Scaling, which combines Dynamic Weight Scaling (DWS) and Attention Stability Norm (ASNorm). DWS allows us to control the scale of values within the attention computation, leading to a more balanced distribution across nodes, similar to the dense weight model. Additionally, we explore the ASNorm to support the stable search further, as detailed in the following sections.

**Dynamic Weight Scaling:** To address the imbalanced attention distribution observed in SLTs, we introduce a Dynamic Weight Scaling (DWS) approach that leverages the magnitude of weight scores during training. In the SLTH, weights are initialized randomly and remain fixed, which limits the adjustment of the magnitude of weights. Consequently, this restriction can lead to worse scaling of activations and hampers SLT exploration. The DWS method uses a factor $\lambda$ based on the absolute mean of the weight scores $\mathbf{S}$ and re-scales the weight matrix $\mathbf{W}_{\mathrm{rand}}$ to produce new $\mathbf{W}_{\mathrm{adp}}$. The process is shown in the following:

$$\mathbf{W}_{\mathrm{adp}} = \frac{1}{mn} \sum_{i=1}^{m} \sum_{j=1}^{n} |\mathbf{S}_{ij}| \cdot \mathbf{W}_{\mathrm{rand}} = \lambda \cdot \mathbf{W}_{\mathrm{rand}}, \tag{6}$$

$$\mathbf{W}_{\mathrm{SLT}} = \mathbf{W}_{\mathrm{adp}} \odot \mathbf{M} = \lambda \cdot \mathbf{W}_{\mathrm{rand}} \odot \mathbf{M}, \tag{7}$$

where $m$ and $n$ are the dimensions of the weight $\mathbf{W}$, score $\mathbf{S}$, and supermask $\mathbf{M}$.

In the Edge-popup algorithm, while weights remain fixed, the weight scores are updated dynamically through gradient descent. The pruning decision relies solely on the relative ranking of scores. However, multiplying the weights with the absolute mean of the weight scores, we incorporate these scaling factors into the learning process, making it more adaptive than using only the relative rankings for pruning.

**Attention Stability Norm:** While DWS contributes to stabilizing attention distributions, the Attention Stability Norm (ASNorm) is necessary to further enhance training stability, particularly in deep-layer models. In our SLT search, we incorporate PairNorm (Zhao & Akoglu, 2020), a normalization technique tailored for GNNs, to control the magnitude of node features within each layer effectively. PairNorm is designed to mitigate over-smoothing in MPNNs. It is achieved by maintaining a constant total pairwise feature distance across layers, which prevents node embeddings from becoming overly similar while preserving meaningful structural information. We also compare different normalization techniques, including BatchNorm (Ioffe

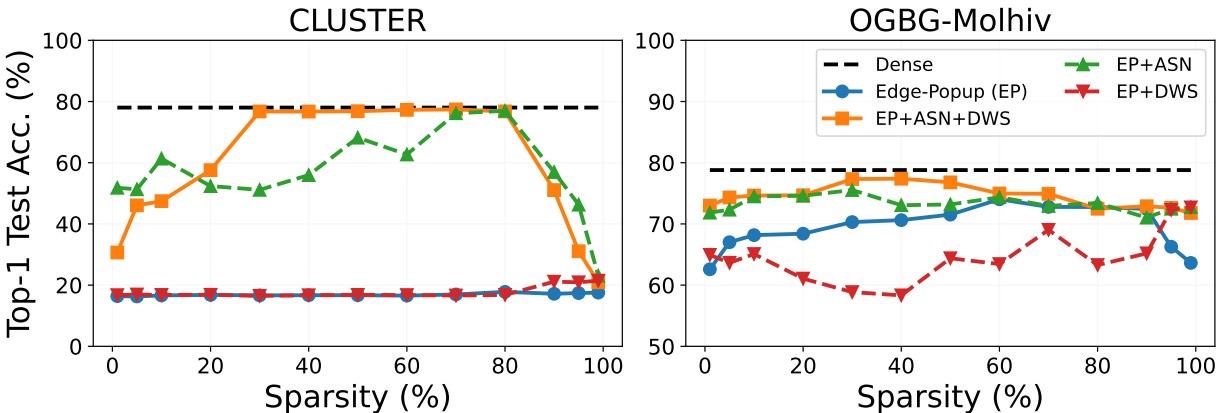

Figure 3: Results comparing Dense and Edge-Popup with Attention Stability Norm, Dynamic Weight Scaling, and both (Adaptive Scaling).

& Szegedy, 2015), LayerNorm (Ba et al., 2016), and RMSNorm (Zhang & Sennrich, 2019), and find that PairNorm consistently provides the most stable and better results for SLT-based GTs in Table1.

Table 1: Comparing accuracy (%) across different normalization methods.

| Normalization | PairNorm | BatchNorm | LayerNorm | RMSNorm | No Norm |
|---|---|---|---|---|---|
| **CLUSTER** | 78.01 | 42.95 | 19.69 | 18.30 | 64.63 |
| **OGBG-Molhiv** | 74.01 | 66.86 | 68.03 | 67.08 | 73.13 |

**Performance of Adaptive Scaling:** To quantitatively assess the contribution of each component, we conducted an ablation study comparing four variants: Edge-Popup (baseline), Edge-Popup + ASNorm, Edge-Popup + DWS, Edge-Popup + DWS + ASNorm (Adaptive Scaling), and Dense.

Figure 3 presents the results of this ablation study. Notably, while DWS alone occasionally decreases accuracy compared to the baseline, the combination of DWS and ASNorm consistently surpasses the baseline. For example, on CLUSTER, Edge-Popup + DWS achieves 73.92%, and Edge-Popup + ASNorm achieves 75.83%, while the combination of both (Adaptive Scaling) reaches 77.41%. These findings underscore the complementary nature of DWS and ASNorm—ASNorm stabilizes attention distributions, enabling DWS to balance weight magnitudes more effectively.

The effect of Adaptive Scaling is further illustrated in Figure 2 (right), which shows how it stabilizes attention distributions. These results highlight the importance of combining DWS and ASNorm to enhance SLT performance.

**Algorithm for Searching SLTs with Adaptive Scaling:** The pseudo-code of our SLT search algorithm is presented in Algorithm 1. It outlines the process for searching SLTs using the Edge-Popup with Adaptive Scaling (DWS and ASNorm).

First, the weight matrix $\mathbf{W}_{\mathrm{rand}}$ and the score matrix $\mathbf{S}$ are randomly initialized using a random seed. Second, the node embeddings $\mathbf{X}$ are normalized using PairNorm to stabilize searching. Third, the algorithm iterates over $T$ epochs to search for the accurate subnetworks with sparsity level $s_T$. In each epoch, the threshold $s_{\mathrm{thres}}$ corresponding to the top $s_T$ fraction of absolute values in $\mathbf{S}$ is computed. A binary supermask $\mathbf{M}$ is generated by setting $M_{ij} = 1$ if $|S_{ij}| \geq S_{\mathrm{thres}}$ and $M_{ij} = 0$ otherwise. DWS is applied by calculating the scaling factor $\lambda = \mathrm{mean}(|\mathbf{S}|)$ and updating the weights as $\mathbf{W}_{\mathrm{adp}} = \lambda \cdot \mathbf{W}_{\mathrm{rand}}$. The sparse model weights are obtained by element-wise multiplying $\mathbf{W}_{\mathrm{adp}}$ with the supermask $\mathbf{M}$, resulting in $\mathbf{W}_{\mathrm{SLT}} = \mathbf{W}_{\mathrm{adp}} \odot \mathbf{M}$. Finally, the score matrix $\mathbf{S}$ is updated using gradient descent on the loss function $\mathcal{L}$ computed with $\mathbf{W}_{\mathrm{SLT}}$.

---

**Algorithm 1:** Searching for SLTs within Graph Transformers using Adaptive Scaling.

---

**Input:** Graph Transformer $g(\mathbf{A}, \mathbf{X}; \mathbf{W}_{\text{rand}})$, weight scores $\mathbf{S}$,
model sparsity $s_T$, learning rate $\gamma$, epochs $T$, ground truth labels $\mathbf{y}$
**Output:** $g(\mathbf{A}, \mathbf{X}_{\text{adp}}, \mathbf{W}_{\text{SLT}})$
Randomly initialize weights $\mathbf{W}_{\text{rand}}$ and scores $\mathbf{S}$ from seed
`# ASNorm: Normalize the Node embeddings`
$\mathbf{X}_{\text{adp}} \leftarrow \text{PairNorm}(\mathbf{X})$
**for** $t = 1$ **to** $T$ **do**
   `# Get the threshold at top` $s_T$ `by sorting` $\mathbf{S}$ `in descending order`
   $k_{\text{th}} \leftarrow \text{int}(\text{len}(\mathbf{S}) \times s_T)$
   $s_{\text{thres}} \leftarrow \text{sort}(|\mathbf{S}|)[k_{\text{th}}]$
   `# Generate the binary mask using` $s_{\text{thres}}$
   **for** *each element* $[i, j]$ *in* $\mathbf{S}$ **do**
      $\mathbf{M}[i, j] \leftarrow 0$ **if** $\mathbf{S}[i, j] < s_{\text{thres}}$ **else** $1$
   `# DWS: scale the magnitude of weight`
   $\lambda \leftarrow \text{mean}(|\mathbf{S}|))$
   $\mathbf{W}_{\text{adp}} \leftarrow \lambda \cdot \mathbf{W}_{\text{rand}}$
   `# Pruning weight by element-wise multiplication with Supermask M`
   $\mathbf{W}_{\text{SLT}} \leftarrow \mathbf{W}_{\text{adp}} \odot \mathbf{M}$
   `# Update` $\mathbf{S}$ `from gradient`
   $\mathbf{S} \leftarrow \mathbf{S} - \gamma \nabla_{\mathbf{S}} \mathcal{L}(g(\mathbf{A}, \mathbf{X}_{\text{adp}}; \mathbf{W}_{\text{SLT}}, \mathbf{y})$
**Return** $g(\mathbf{A}, \mathbf{X}_{\text{adp}}, \mathbf{W}_{\text{SLT}})$

---

In this study, we search for SLTs by applying supermask to the weights, as shown in Equation (5). We have modified the linear projection components within the GT layers, which are the primary layers utilizing the self-attention mechanism. In other words, we can search for SLTs without modifying the attention mechanisms, making them flexible enough to fit any GT architecture. We apply the supermask to all linear projection layers in the Transformer layer, not only to the query, key, and value matrix of the self-attention but also to the weights in the feed-forward networks.

## 4 Experiments and Results

This section describes experiments using several Graph Transformers (GTs) architectures and datasets to assess the performance of the SLTs. We analyze the effectiveness, particularly in the robustness toward perturbations and out-of-distribution (OOD) detection, which are considered challenging in GTs. We also show memory reduction of SLTs compared with dense weight learning models.

**Baseline Architectures:** We use three popular models to validate that SLTs exist in versatile GT architectures. The first is NodeFormer (Wu et al., 2022a), a pure GT designed for node-level classification in large graphs. The second one, GRIT (Ma et al., 2023), is a pure GT for graph-level tasks. The third one, GraphGPS (Rampášek et al., 2022), is a hybrid GT that combines both local message passing and global attention mechanisms for graph-level tasks. Our experiments follow the configurations (the number of layers, etc.) and hyperparameters of their original papers, adjusting the learning rate and SLT sparsity.

**Datasets:** We use the same datasets as the original papers for fair comparison. For NodeFormer, we use seven tasks from Cora, Citeseer, Pubmed (Sen et al., 2008), Deezer (Rozemberczki & Sarkar, 2020), Actor (Pei et al., 2020), Mini-ImageNet (Vinyals et al., 2016) and 20News-Groups (Pedregosa et al., 2011). Cora, Citeseer, and Pubmed are citation networks, Deezer and Actor are social networks, Mini-ImageNet is an image classification, and 20News-Groups is a text classification task. For GRIT and GraphGPS, we use five tasks, ZINC, MNIST, CIFAR10, PATTERN, and CLUSTER, from Benchmarking GNNs (Dwivedi et al., 2023), and OGBG-Molhiv from Open Graph Benchmark (OGB) (Hu et al., 2020). ZINC is the molecular regression task, MNIST and CIFAR10 are the image classification tasks, and PATTERN and CLUSTER

are synthetic datasets sampled from the stochastic block model. OGBG-Molhiv is designed for molecular property prediction. The details of the datasets used are provided in Appendix A provides the details of the datasets used.

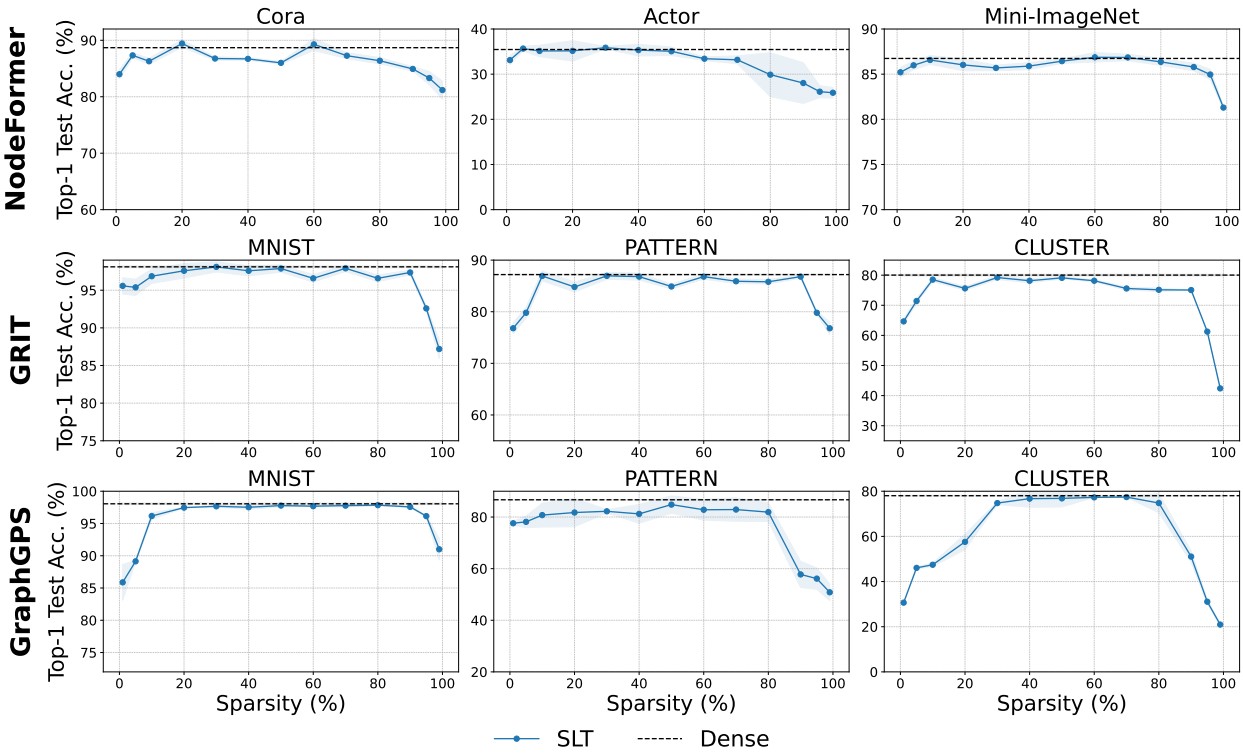

Figure 4: Comparison of the accuracy of SLTs and dense weight learning models across varying sparsity levels in different GT architectures and datasets. The top row corresponds to NodeFormer evaluated on the Cora, Actor, and Mini-ImageNet datasets. The middle row presents GRIT's performance on MNIST, PATTERN, and CLUSTER datasets, while the bottom row shows GraphGPS evaluated on the same datasets. The results demonstrate that SLTs achieve accuracy comparable to dense models across a wide range of sparsity levels (10% to 90%).

## 4.1 Accuracy Benchmarking on SLTs

In Figure 4, we plot accuracy against sparsity (1% ∼ 99%) for SLTs in the three architectures. In the NodeFormer on Cora, the dense model achieves an accuracy of 88.69%, while the SLT with 60% sparsity attains a higher accuracy of 89.42%. At very low (<10%) and very high (>90%) sparsity levels, accuracy drops considerably (e.g., accuracy decreases to 81.18% at 99% sparsity on Cora). Across these architectures, we observe that SLT peaks within a moderate sparsity range (10% to 90%), aligning closely with or surpassing dense models' performance.

## 4.2 Accuracy-to-Memory Size Trade-Off

Given their high computational and memory demands, memory efficiency is a critical challenge for GTs. Figure 5 further illustrates the significant improvements in the trade-off between accuracy and memory. SLTs consistently achieve similar or superior accuracy, reducing memory demands compared to the dense models. This steep improvement at lower memory levels demonstrates how SLTs enable GTs to perform effectively even in highly resource-constrained environments, making them well-suited for deployment on edge devices or other memory-limited systems.

Table 2 compares NodeFormer-SLT with NodeFormer-Dense, traditional MPNN models GCN and GAT. It shows that NodeFormer-SLT achieves the highest accuracy (78.20%) while consuming only 0.619 MB of memory—significantly less than NodeFormer-Dense (4.512 MB), GCN (0.982 MB), or GAT (0.982 MB). These are all two-layer architectures. This demonstrates the effectiveness of SLTs in drastically reducing memory usage while maintaining or improving model performance compared to dense GTs and MPNNs.

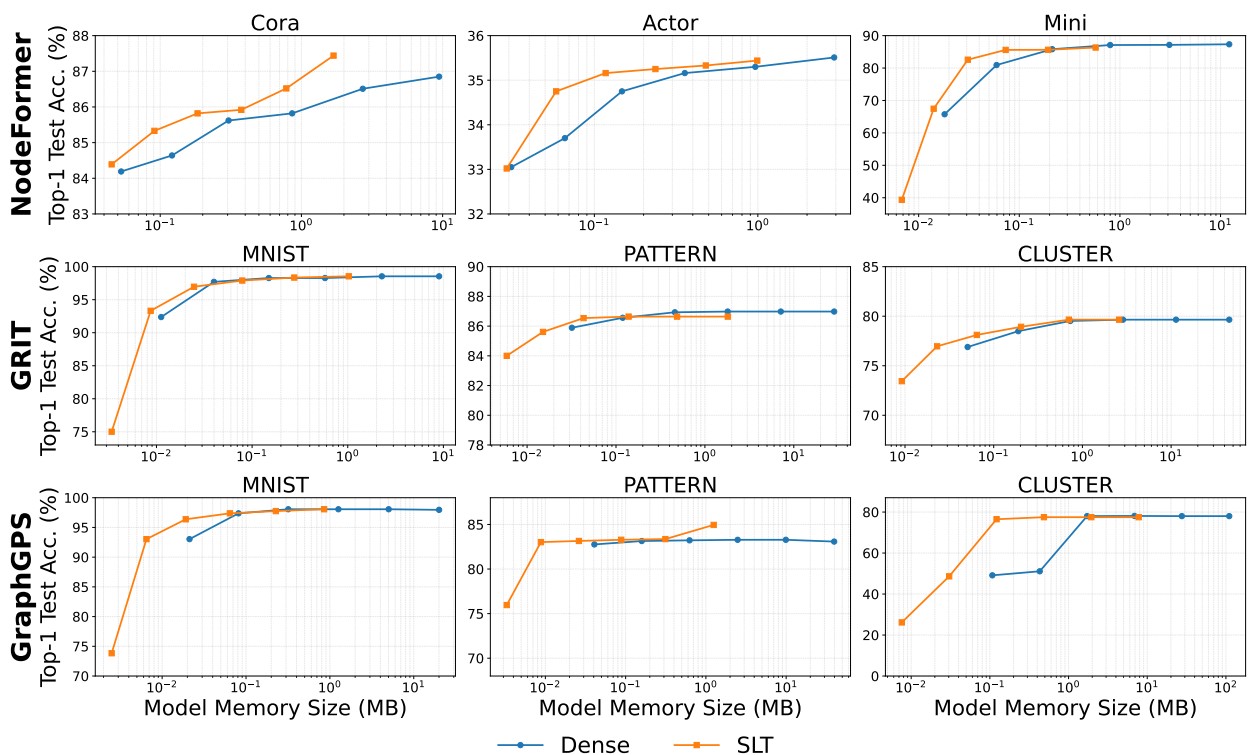

Figure 5: Comparison of the trade-off between model memory usage and accuracy for SLT and dense weight learning models. The hidden channels are varied across values of 8, 16, 32, 64, 128, and 256. This figure illustrates that the SLT model significantly improves the trade-off, achieving high accuracy while substantially reducing memory consumption compared to the dense model.

Table 2: Comparison of memory usage and accuracy across different models on Pubmed datasets (2-layer, 256 hidden channels) with NodeFormer (SLT, Dense), GCN, and GAT.

|  | NodeFormer-SLT | NodeFormer-Dense | GCN | GAT |
|---|---|---|---|---|
| **Memory (MB)** | 0.619 (×7.3 ↓) | 4.512 | 0.982 | 0.982 |
| **Accuracy (%)** | 78.20 | 75.54 | 75.98 | 76.4 |

## 4.3 Out-of-Distribution (OOD) Data Detection

In this section, we compare the generalization of SLTs in GTs by evaluating the ability to detect and manage OOD instances against dense models. A key indicator of model robustness, particularly in real-world scenarios, is its performance on OOD data, *i.e.* data that present patterns or characteristics unseen during training. It is often critical for real-world applications of GTs, as they can process graphs with new or unexpected structures, potentially leading to performance degradation if the model overfits the training data distributions. Our experiments utilize OOD test sets across various tasks to evaluate to what extent SLTs can generalize beyond the training data distribution.

To evaluate OOD detection performance, we used ROC-AUC as the primary metric. As shown in Figure 6, SLTs respond more robustly to OOD data, achieving higher detection rates and better classification stability than dense models. For instance, SLTs improve accuracy by approximately 0.15 to 0.4 on the Cora dataset, 0.2 to 0.3 on Citeseer, and 0.05 to 0.2 on Actor. These gains likely stem from the inherent regularization effects of SLTs, which help mitigate overfitting and maintain performance on unseen data distributions. By enforcing sparsity, SLTs focus on the most salient features and connections, improving generalization to OOD data. Consequently, SLTs enhance OOD robustness, improving model reliability and safety in critical applications.

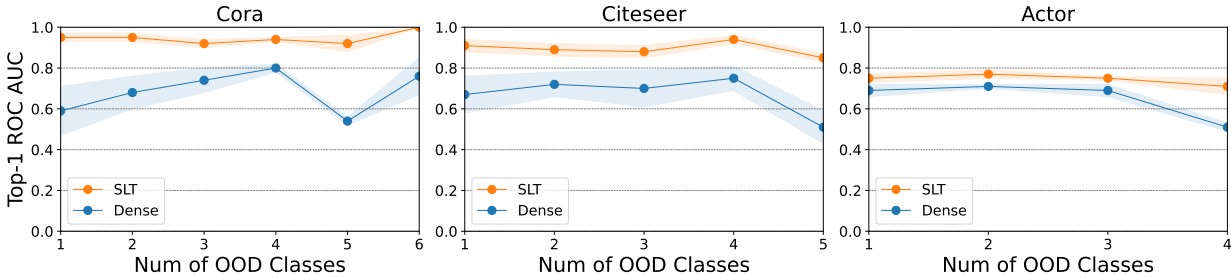

Figure 6: Performance comparison on OOD data between SLT and dense models, evaluated by ROC-AUC. The x-axis represents the number of OOD classes randomly selected from the dataset. During training, OOD classes were removed, and the models were tested on their ability to detect these unseen classes.

## 4.4 Perturbation

Real-world graph data often contains noise or errors, making robustness to perturbations critical for reliable performance. Perturbations typically occur in two forms: node perturbations, where noise is added to node features (e.g., due to sensor inaccuracies or incomplete information), and edge perturbations, which modify graph connections to simulate errors or dynamic changes in relationships. By assessing both types, we evaluate the robustness of SLTs compared to dense models, highlighting their ability to handle noisy inputs and structural modifications effectively.

**Node Perturbation:** We examine the robustness of SLTs under node perturbations by adding Gaussian noise to node features during inference and simulating common real-world scenarios such as noisy sensor data or incomplete information in social networks.

The perturbation is applied to each node feature by adding Gaussian noise with strength controlled by a factor $\epsilon$. The perturbed feature $\tilde{x}$ is given by:

$$\tilde{x} = x + z, \quad z \sim \mathcal{N}(0, \epsilon^2), \tag{8}$$

where $x$ is the element of original node feature, and $z$ is sampled from a standard normal distribution. Since $z$ scales the noise by $\epsilon$, $\tilde{x}$ retains a Gaussian distribution with mean 0 and variance $\epsilon^2$.

Figure 7 indicate that SLTs outperform dense models even under high noise levels, maintaining stable accuracy as $\epsilon$ increases to a certain threshold. Specifically, on the Cora dataset, SLTs achieve a 1% to 5% improvement when $\epsilon$ exceeds 0.04. For Citeseer, SLTs show around a 5% improvement starting from $\epsilon = 0.02$. On Pubmed, SLTs demonstrate robustness even at small $\epsilon$, with a gradual increase in performance and up to a 10% improvement under higher noise levels. This robustness suggests that the sparsity introduced in SLTs acts as a form of regularization, reducing model complexity and preventing overfitting to noise-free training data. By focusing on the most critical connections, SLTs may enhance the ability to generalize to perturbed inputs, improving robustness against noise.

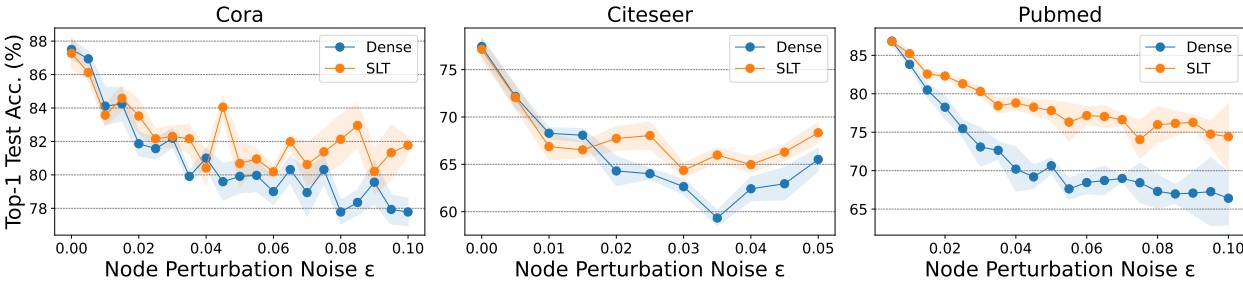

Figure 7: Classification accuracy of SLTs and dense models under varying levels of node feature perturbation on the Cora, Citeseer, and Pubmed datasets. Gaussian noise with standard deviation $\epsilon$ is added to the node features to simulate node perturbation.

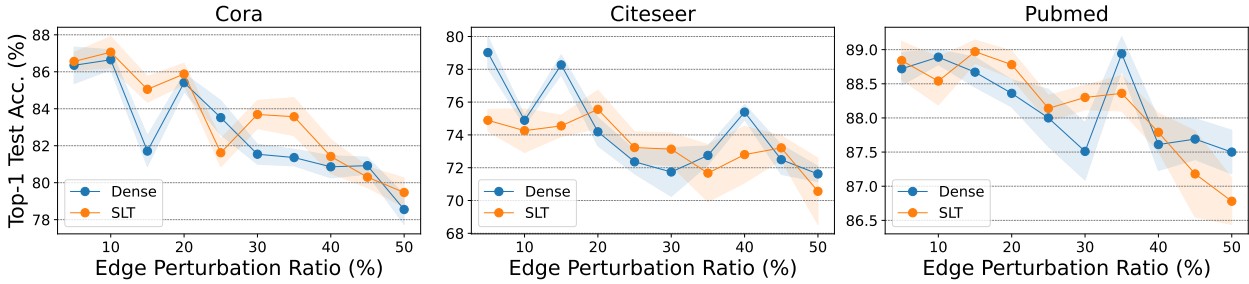

Figure 8: Classification accuracy of SLTs and fully trained dense models under varying levels of edge perturbation on benchmark datasets. Edge perturbation is performed by randomly reassigning a certain percentage of edges to different destination nodes.

**Edge Perturbation:** In addition to node feature perturbation, we test the robustness of SLTs under edge perturbation. This process simulates real-world situations where relational data might be incomplete or contain erroneous connections, such as in dynamic or evolving networks. The edge perturbation process involves selecting a subset of edges and replacing each edge's destination node with a random alternative node during inference, controlled by a parameter representing the percentage of edges modified.

As shown in Figure 8, SLTs do not exhibit a distinct advantage over dense models under edge perturbation. This result stems from a key characteristic of GTs: unlike MPNN, they do not rely on direct edge connections for information aggregation. Instead, graph structure, including edges, is often encoded as part of the node features in the preprocessing stage or used as a supplementary factor in the aggregation process. In other words, SLTs have the least effect on edges or graph structure. Therefore, while SLTs exhibit significant robustness under node perturbation, they do not confer a comparable robustness advantage under structural modifications like edge perturbation.

# 5   Limitations and Future Works

Our proposed SLT approach has shown considerable benefits in reducing memory consumption and improving robustness in GTs. However, there are several limitations and promising directions for future research.

**Extension to a Broader Range of Applications:** While we have demonstrated the effectiveness of SLTs in GTs for node-level and graph-level tasks, extending our findings to a wider variety of graph types and applications—such as link prediction and graph generation—would help further validate the versatility and scalability of our method.

**Large-Scale Graphs:** In extremely large-scale graphs, techniques such as sparse attention mechanisms or graph partitioning are commonly used. In these cases, the global scope of the attention mechanism

is explicitly reduced, as the computation that originally considered all node pairs may be limited to a smaller subset. Investigating the combination of SLT with these techniques, particularly when the perceptual scope decreases, will help us understand whether SLT can still be effective. This can lead to more efficient acceleration for large-scale graphs.

**Unstructured or Structured Pruning:** Our method, as well as most other SLTH-based approaches, employs an unstructured pruning approach. However, structured pruning is often preferred in hardware like GPUs, as it can improve resource utilization and, thus, the inference speed, while it potentially impacts accuracy. It is worth exploring the trade-off between unstructured and structured pruning with constraints on accuracy and memory size.

**Theoretical Insights Into SLTs for GTs:** While this study primarily focuses on empirical validation, the theoretical underpinnings of SLTs in the context of GTs remain underexplored. Bridging this gap could provide a deeper understanding of why SLTs emerge in GTs and how their properties differ from those in traditional neural networks or general Transformers. We encourage future research to develop a more formal theoretical framework to support our findings.

## 6 Conclusion

In this paper, we explored the existence of SLTs within GTs. We highlighted their effectiveness in addressing key challenges, such as high memory consumption, lack of robustness to perturbations, and limited generalization to OOD data.

- We provide the first empirical evidence of the existence of SLTs within GTs, demonstrating their applicability across both pure and hybrid GT architectures.

- We identify that traditional SLT methods can fail to find SLTs within GTs, so we introduce Adaptive Scaling, a novel technique that balances attention distributions and improves search stability.

- Our findings show that SLTs significantly reduce memory consumption, achieving up to $32\times$ reduction while maintaining or even outperforming the accuracy of fully trained dense models. Furthermore, the proposed SLT method enhances robustness against adversarial perturbations and generalizations to OOD data, ensuring model reliability in real-world scenarios.

## Acknowledgments

This work was supported in part by JST PRESTO Grant Number JPMJPR22P7, JSPS KAKENHI Grant Number JP23H05489, and JST-ALCA-Next Japan Grant # JPMJAN24F3.

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

# Appendix

## A  Datasets

For node-level prediction, We use seven datasets: Cora, Citeseer, Pubmed, Deezer, Actor, Mini-ImageNet, and 20News-Groups. The Cora, Citeseer, and Pubmed datasets are citation network datasets where each document is represented as a node, and each citation link between documents serves as an edge. Deezer is a social network of users from European countries on the Deezer platform, where edges show mutual follower relationships. The features for each node come from the artists each user likes, and nodes are labeled with gender. The Actor dataset is a graph of actors appearing together on Wikipedia pages. Each node is an actor, and an edge between two nodes means both actors were mentioned on the same Wikipedia page. The 20News-Groups dataset contains around 20,000 documents from newsgroups, organized (almost) equally across 20 different groups, with each document represented as a node. The Mini-ImageNet dataset comprises $84 \times 84$ RGB images, covering 100 classes with 600 images in each class.

We use six datasets for graph-level prediction: ZINC, MNIST, CIFAR10, PATTERN, CLUSTER, and OGBG-Molhiv. ZINC is 12,000 molecular graphs from the ZINC database, tasked with regressing constrained solubility (logP) values. CIFAR10 and MNIST are image datasets from SLIC superpixels from 8-nearest-neighbor graphs for 10-class classification. PATTERN is a synthetic dataset using Stochastic Block Models to classify nodes into one of 100 subgraph patterns. CLUSTER is a synthetic dataset of graphs with six SBM-generated clusters, aiming to classify nodes by cluster-ID. OGBG-Molhiv is a molecular property dataset for binary classification of a molecule's ability to inhibit HIV replication.

Table 3: Overview of the graph dataset used in this study.

| DataSets | #Graphs | Avg. #Nodes | Avg. #Edges | #Node Feat | #Classes | Metric |
|---|---|---|---|---|---|---|
| Cora | 1 | 2,708 | 5,429 | 1,433 | 7 | Accuracy |
| Citeseer | 1 | 3,327 | 4,732 | 3,703 | 6 | Accuracy |
| Pubmed | 1 | 19,717 | 88,648 | 500 | 3 | Accuracy |
| Deezer | 1 | 28,281 | 92,752 | 31,241 | 2 | ROC-AUC |
| Actor | 1 | 7,600 | 29,926 | 931 | 5 | Accuracy |
| Mini-ImageNet | 1 | 18,000 | 0 | 128 | 30 | Accuracy |
| 20News-Groups | 1 | 9,607 | 0 | 236 | 10 | Accuracy |
| ZINC | 12,000 | 23.2 | 24.9 | 1 | 0 | MAE |
| MNIST | 70,000 | 70.6 | 564.5 | 3 | 10 | Accuracy |
| CIFAR10 | 60,000 | 117.6 | 941.1 | 5 | 10 | Accuracy |
| PATTERN | 14,000 | 118.9 | 3,039.3 | 3 | 2 | Accuracy |
| CLUSTER | 12,000 | 117.2 | 2,150.9 | 7 | 6 | Accuracy |
| OGBG-Molhiv | 41,127 | 25.5 | 27.5 | 9 | 2 | ROC-AUC |

## B  Supplementary Experimental Results

Experiments evaluate for NodeFormer on Cora, Citeseer, Pubmed, Deezer, Actor, Mini-ImageNet, and 20News-Groups, for GRIT and GraphGPS on ZINC, MNIST, CIFAR10, PATTERN, CLUSTER, and OGBG-Molhiv. We compare SLT performance with the fully trained dense model in Table 4 5.

Table 4: Experiments comparison accuracy between fully trained dense model and SLT of NodeFormer on six benchmark datasets.

| Datasets | Cora | Citeseer | Deezer | Actor | Mini-ImageNet | 20News-Groups |
|---|---|---|---|---|---|---|
| NodeFormer-Dense | $88.69 \pm 0.46$ | $76.33 \pm 0.59$ | $71.24 \pm 0.32$ | $35.31 \pm 1.29$ | $86.74 \pm 0.23$ | $65.21 \pm 1.14$ |
| NodeFormer-SLT | $89.42 \pm 0.56$ | $76.93 \pm 1.30$ | $72.96 \pm 0.48$ | $37.62 \pm 0.40$ | $87.21 \pm 0.54$ | $65.60 \pm 0.38$ |

Table 5: Experiments comparison accuracy between fully trained dense model and SLT of GRIT and GraphGPS on five benchmark datasets.

| Datasets | ZINC | MNIST | CIFAR10 | PATTERN | CLUSTER |
|---|---|---|---|---|---|
| | MAE $\downarrow$ | Accuracy $\uparrow$ | Accuracy $\uparrow$ | Accuracy $\uparrow$ | Accuracy $\uparrow$ |
| GRIT-Dense | $0.059 \pm 0.002$ | $98.10 \pm 0.111$ | $76.468 \pm 0.881$ | $87.196 \pm 0.076$ | $80.026 \pm 0.277$ |
| GRIT-SLT | $0.077 \pm 0.005$ | $98.10 \pm 0.324$ | $71.36 \pm 0.37$ | $86.95 \pm 0.51$ | $79.22 \pm 0.42$ |
| GraphGPS-Dense | $0.070 \pm 0.004$ | $98.051 \pm 0.126$ | $72.298 \pm 0.356$ | $86.685 \pm 0.059$ | $78.016 \pm 0.180$ |
| GraphGPS-SLT | $0.098 \pm 0.014$ | $97.980 \pm 0.153$ | $72.460 \pm 0.422$ | $83.941 \pm 0.272$ | $77.501 \pm 0.251$ |

The effectiveness of SLTs depends on dataset characteristics. SLTs excel in node classification tasks (Table 4) due to local regularities and redundancies in node features, which are well-suited to subnetworks. However, they underperform in graph classification tasks (Table 5), which require modeling global structural features from diverse and irregular graphs—demands that often exceed the capacity of fixed subnetworks.

SLTs perform better on simpler datasets with fewer nodes and edges but face challenges with larger, more complex datasets like CIFAR10, where denser connectivity is needed to capture global information. These results suggest that SLTs are more effective for tasks with strong local structures and lower global complexity, while dense models remain crucial for tasks requiring robust global representations.

## C  Comprehensive Hyperparameter Settings

This section provides comprehensive hyperparameter settings to ensure the reproducibility of our experiments. Tables 6, 7, and 8 detail the configurations used for different models and datasets.

Table 6: NodeFormer-SLT hyperparameters for six datasets.

| Hyperparameter | Cora | CiteSeer | Deezer | Actor | Mini-ImageNet | 20News-Group |
|---|---|---|---|---|---|---|
| # Transformer Layers | 2 | 2 | 2 | 2 | 2 | 2 |
| Hidden dim | 32 | 32 | 32 | 32 | 128 | 64 |
| # Heads | 4 | 2 | 1 | 1 | 6 | 4 |
| Learning rate | 0.007 | 0.003 | 2e-4 | 0.007 | 0.001 | 0.001 |
| Weight decay | 5e-3 | 5e-3 | 5e-2 | 5e-2 | 5e-3 | 5e-3 |
| # Epochs | 1000 | 1000 | 1000 | 1000 | 300 | 200 |

## D  Comparison to Additional Pruning Baselines

This section provides a detailed comparison of our proposed SLT approach with additional pruning baselines. We conducted experiments with two representative pruning methods: N:M Sparsity (Zhou et al., 2021a) and BitNet (Ma et al., 2024). These methods were chosen for their relevance in structured pruning and Transformer-specific pruning techniques, respectively. Below, we summarize the methodologies and results of these experiments.

Table 7: GraphGPS-SLT hyperparameters for six datasets.

| Hyperparameter | ZINC | MNIST | CIFAR10 | PATTERN | CLUSTER | OGBG-Molhiv |
|---|---|---|---|---|---|---|
| # GPS Layers | 10 | 3 | 3 | 6 | 16 | 10 |
| Hidden dim | 64 | 52 | 52 | 64 | 48 | 64 |
| # Heads | 4 | 4 | 4 | 4 | 8 | 4 |
| Batch size | 32 | 16 | 16 | 64 | 16 | 32 |
| Learning rate | 0.001 | 0.001 | 0.001 | 0.0005 | 0.0005 | 0.0001 |
| Weight decay | 1e-5 | 1e-5 | 1e-5 | 1e-5 | 1e-5 | 1e-5 |
| # Epochs | 2000 | 100 | 100 | 100 | 100 | 100 |

Table 8: GRIT-SLT Hyperparameters for six datasets.

| Hyperparameter | ZINC | MNIST | CIFAR10 | PATTERN | CLUSTER | OGBG-Molhiv |
|---|---|---|---|---|---|---|
| # Transformer Layers | 10 | 3 | 3 | 10 | 16 | 10 |
| Hidden dim | 64 | 52 | 52 | 64 | 48 | 64 |
| # Heads | 8 | 4 | 4 | 8 | 4 | 4 |
| Batch size | 32 | 16 | 16 | 32 | 16 | 32 |
| Learning Rate | 0.001 | 0.001 | 0.001 | 0.0005 | 0.0005 | 0.0001 |
| Weight decay | 1e-5 | 1e-5 | 1e-5 | 1e-5 | 1e-5 | 1e-5 |
| # Epochs | 2000 | 200 | 200 | 100 | 100 | 100 |

- **N:M Sparsity**: This structured pruning approach enforces a fixed ratio of non-zero weights in each block of parameters, effectively reducing model complexity while maintaining a predefined sparsity level. N:M Sparsity has been widely adopted for efficient Neural Network architectures.

- **BitNet**: A pruning and quantization method designed specifically for Transformers, BitNet leverages weight binarization to reduce memory footprint while preserving task performance. It has demonstrated effectiveness in compressing large Transformer models with minimal degradation in accuracy.

We used NodeFormer as a benchmark model across the same datasets and evaluation metrics as in our main experiments. The comparison results are presented in Table 9.

Table 9: Comparison of pruning methods.

| Method | Accuracy (%) | OOD Performance (%) | Node Perturbation Performance (%) |
|---|---|---|---|
| **N:M Sparsity** | 85.77 | 80.97 | 78.43 |
| **BitNet** | 84.49 | 81.79 | 78.88 |
| **Ours (SLT-based)** | **89.42** | **92.01** | **82.13** |

The results demonstrate that our SLT-based approach significantly outperforms both N:M Sparsity and BitNet in accuracy, out-of-distribution (OOD) performance, and robustness under node perturbations. Our method achieves the highest accuracy (89.42%), a notable improvement over N:M Sparsity (85.77%) and BitNet (84.49%). This reflects the ability of SLTs to retain critical parameters while pruning unimportant ones effectively. The OOD performance of our method (92.01%) exceeds that of N:M Sparsity (80.97%) and BitNet (81.79%), highlighting its robustness in generalizing to unseen data distributions. Under node perturbation scenarios, our method maintains a superior performance (82.13%), outperforming N:M Sparsity (78.43%) and BitNet (78.88%). These findings validate the efficacy of our SLT-based approach in addressing the challenges of pruning in GTs. Furthermore, this comparison establishes the competitiveness of our method not only against Edge-Popup but also other advanced pruning techniques.

# E    Comparisons with Existing SLTs Methods in GNNs

In this section, we provide a detailed comparison of our proposed SLT approach with existing SLT research on GNNs. Specifically, we compare our method with UGTs Huang et al. (2022) and MF-GNN Yan et al. (2024), which have studied SLTs in MPNNs such as GCNs.

- **UGTs**: UGTs employ a single mask and do not take the absolute value of the weight scores during mask generation. It can be considered a variant of Edge-Popup in the context of GNNs.

- **MF-GNN**: MF-GNN merges multicoated supermasks (MSup) into a single combined mask to identify high-performing SLTs.

We first review the results reported in their respective studies using GCN. Subsequently, we adapt their methods to the GT setting (NodeFormer) by employing the same dataset splits used in their original experiments.

Table 10 presents the accuracy comparison across GCN and NodeFormer.

Table 10: Comparison of our SLT-based method with existing SLT methods in GNNs, adapted to the Graph Transformer setting (NodeFormer).

| Method | GCN Accuracy (%) | NodeFormer Accuracy (%) |
|---|---|---|
| UGTs | 77.03 | 77.13 |
| MSup (MF-GNN) | 80.80 | 73.20 |
| Ours (SLT-based) | – | **81.23** |

The results highlight several key observations. On GCNs, MSup (MF-GNN) achieves the highest accuracy (80.80%), followed by UGTs (77.03%). In both methods on GT setting, UGTs achieve 77.13%, and MSup (MF-GNN) drops to 73.20%. Our SLT-based approach outperforms both methods in the NodeFormer setting, achieving 81.23%, demonstrating the effectiveness of our proposed Adaptive Scaling techniques in finding SLTs within GTs.

Our findings suggest that while existing SLT methods excel in MPNN-based models, they face limitations when applied to Transformer-based architectures, further emphasizing the contributions of our proposed techniques.

# F    Hyperparameter Sensitivity Analysis

In this section, we analyze the sensitivity of our proposed SLT-based approach to hyperparameter variations. Specifically, using the Cora dataset, we examine the effect of three critical hyperparameters: learning rate, number of layers, and hidden dimensions. This analysis complements the sparsity-level results in Figure 4.

## F.1    Learning Rate

Table 11 summarizes the results of varying the learning rate across a range of values from 0.0001 to 0.05. The comparison includes both the Dense baseline and our SLT.

Table 11: Comparison of Dense and SLT with different learning rates.

| Method | 0.0001 | 0.0005 | 0.001 | 0.005 | 0.007 | 0.01 | 0.05 |
|---|---|---|---|---|---|---|---|
| Dense | 84.88 | 85.43 | 85.77 | 86.31 | 86.07 | 85.82 | 36.78 |
| SLT | 79.96 | 84.54 | 85.52 | 86.66 | 86.76 | 86.61 | 44.90 |

As shown in Table 11, performance declines significantly when the learning rate deviates far from the optimal range. However, within a broad range of 0.0005 to 0.01, there is no substantial difference in accuracy between the Dense and SLT methods.

## F.2 Number of Layers

We also evaluate the sensitivity of our method to the number of layers in the network. Table 12 presents the results for Dense and SLT across configurations with two to six layers.

Table 12: Comparison of Dense and SLT across the number of layers.

| #Layers | 2 | 3 | 4 | 5 | 6 |
|---------|-------|-------|-------|-------|-------|
| Dense | 85.77 | 85.57 | 85.82 | 85.92 | 86.26 |
| SLT | 86.76 | 85.82 | 60.07 | 36.48 | 34.71 |

The results in Table 12 reveal that while Dense methods exhibit stable performance across different numbers of layers, the accuracy of SLT methods declines sharply beyond four layers. This suggests that the effectiveness of SLT is best realized with an appropriate choice of layer depth, making selecting the number of layers a critical consideration for practical implementations.

## F.3 Hidden Dimensions

We also evaluate the sensitivity of our method to the hidden dimension in the network. Table 13 presents the results for Dense and SLT across configurations with 8 to 256 hidden dimensions in two layers of GT.

Table 13: Comparison of Dense and SLT across hidden dimensions.

| Hidden dimensions | 8 | 16 | 32 | 64 | 128 | 256 |
|-------------------|-------|-------|-------|-------|-------|-------|
| Dense | 84.19 | 85.33 | 85.62 | 85.82 | 86.85 | 86.51 |
| SLT | 84.64 | 85.92 | 86.12 | 87.44 | 85.82 | 84.39 |

Table 13 demonstrates that SLT maintains competitive performance with Dense models even at smaller hidden dimensions, achieving 84.64% accuracy with only eight dimensions compared to 84.19% for Dense. This highlights the efficiency and representational capacity of SLT in low-dimensional settings. Dense methods benefit from increasing hidden dimensions; SLT peaks at 64 dimensions (87.44%), showcasing its ability to leverage sparse representations effectively.

