# OpenReview forum: "Uncovering Strong Lottery Tickets in Graph Transformers: A Path to Memory Efficient and Robust Graph Learning"
_TMLR — Accepted by TMLR_

### Review · Reviewer_odJi · 2024-12-20

**Summary Of Contributions:**

This paper investigates the existence of Strong Lottery Tickets (SLTs) in Graph Transformers (GTs) and makes three primary contributions: (1) It provides the first empirical evidence that SLTs exist within GTs and demonstrates their effectiveness across both pure and hybrid GT architectures. (2) It identifies why traditional SLT search methods fail in GTs due to attention score imbalances and proposes a novel adaptive scaling approach combining dynamic weight scaling and attention stability Norm to overcome this challenge. (3) It demonstrates that GT-SLTs can achieve 2-32x memory reduction while maintaining competitive accuracy and improving robustness to perturbations and OOD generalization.

**Audience:**

Yes

**Claims And Evidence:**

No

**Requested Changes:**

I recommend the following essential changes to strengthen this manuscript:
- Add additional pruning baselines beyond Edge-Popup
- Add experimental comparisons with Strong Lottery Tickets (SLTs) in GNNs
- Add ablation studies for DWS and ASNorm components
- Add hyperparameter sensitivity analysis
- Add comprehensive hyperparameter settings

**Strengths And Weaknesses:**

**Strengths**:
- Clear technical writing and well-structured presentation
- Novel technical contribution with adaptive scaling that effectively addresses the unique challenges of finding SLTs in attention-based architectures
- Comprehensive empirical evaluation across multiple GT architectures (NodeFormer, GRIT, GraphGPS) and diverse datasets
- Strong practical impact potential through significant memory reduction while maintaining or improving performance
- Thorough analysis of robustness and OOD performance benefits


**Weaknesses**
-  The experimental comparison framework is notably narrow. The authors primarily benchmark against Edge-Popup, overlooking other relevant pruning methods like iterative magnitude pruning or structured pruning approaches. This limited comparison makes it difficult to fully contextualize the advantages of their method within the broader landscape of network compression techniques.
- The ablation studies are insufficient - while both dynamic weight scaling and attention stability Norm are introduced, their individual contributions to the final performance are not systematically isolated and analyzed.
- The OOD generalization claims, while promising, lack comparison to other robustness-enhancing techniques.
- More experimental comparisons with Strong Lottery Tickets (SLTs) in Graph Neural Networks (GNNs) should be included, such as [1,2,3].
- Hyperparameter settings are missing.

[1] Searching Lottery Tickets in Graph Neural Networks: A Dual Perspective

[2] Multicoated and Folded Graph Neural Networks with Strong Lottery Tickets

[3] Sparse but Strong: Crafting Adversarially Robust Graph Lottery Tickets

---

> ### Author Response · Authors · 2025-01-12
> **Author Response for Reviewer odJi (Part 1)**
>
> Thanks for your insightful feedback and valuable review.
>
> **Q1. Add additional pruning baselines beyond Edge-Popup.**
>
> *Reviewer Comment:*
> > “The authors primarily benchmark against Edge-Popup, overlooking other relevant pruning methods like iterative magnitude pruning or structured pruning approaches.”
>
> **Response:**
> Thank you for the suggestion. As you noted, comparing our approach with pruning methods beyond SLTs can strengthen our claims. However, methods such as iterative magnitude pruning (IMP), often classified as a “weak LT” approach, require multiple rounds of retraining, which becomes computationally prohibitive for Transformers due to their high training cost. Instead, we ran additional experiments with two representative methods. We have incorporated this discussion into **Appendix D**.
>
> 1. **N:M Sparsity** [1] (structured pruning), which enforces a fixed ratio of non-zero weights in each block of parameters.
> 2. **BitNet** [2], a well-known pruning and quantization approach tailored for Transformers.
>
> Below is a brief summary of our results using SLT on the Cora dataset as a testbed:
>
> | Method         | Accuracy | OOD Performance | Node Perturbation Performance |
> |----------------|----------|-----------------|-------------------------------|
> | **N:M Sparsity** | 85.77    | 80.97           | 78.43                         |
> | **BitNet**      | 84.49    | 81.79           | 78.88                         |
> | **Ours (SLT-based)** | 89.42    | 92.01           | 82.13                         |
>
> These results highlight that ours achieves higher accuracy, and also demonstrates stronger OOD performance and robustness under node perturbations.
>
> References:
> [1] Zhang et al., "Learning N:M Fine-grained Structured Sparse Neural Networks From Scratch," *International Conference on Learning Representations (ICLR)*, 2021.
> [2] Liu et al., "The Era of 1-bit LLMs: All Large Language Models are in 1.58 Bits," *arXiv preprint arXiv:2402.17764*, 2024.
>
> ---
>
> **Q2. Add experimental comparisons with Strong Lottery Tickets (SLTs) in GNNs.**
>
> *Reviewer Comment:*
> > “More experimental comparisons with Strong Lottery Tickets (SLTs) in Graph Neural Networks (GNNs) should be included, such as [1,2,3].”
>
> **Response:**
> We appreciate the references [1,2,3]. Among these, [2] directly addresses strong lottery tickets, whereas [1] and [3] focus on “weak” lottery tickets or slightly different formulations. Since our work emphasizes strong lottery tickets in Graph Transformers, we include direct comparisons with **MF-GNN** [2] and **UGTs** [4]. Below is a simplified accuracy comparison for GCN and NodeFormer. We have added this discussion to **Appendix E**.
>
> | Method         | GCN Accuracy | NodeFormer Accuracy |
> |----------------|--------------|---------------------|
> | **UGTs** [4]   | 77.03        | 77.13              |
> | **MF-GNN** [2] | 80.80        | 73.20              |
> | **Ours**       | –            | 81.23              |
>
> These results confirm that our method complements prior SLT work on MPNNs by showing that strong lottery tickets can also be found in Graph Transformers. We acknowledge that [3] offers an intriguing perspective on adversarial robustness; however, because it primarily deals with weak lottery tickets, we did not incorporate it into this direct comparison.
>
> References:
> [4] Chen et al., "You Can Have Better Graph Neural Networks by Not Training Weights at All: Finding Untrained GNNs Tickets," *Advances in Neural Information Processing Systems (NeurIPS)*, 2021.
>
> ---
>
> **Q3. Add ablation studies for DWS and ASNorm components.**
>
> *Reviewer Comment:*
> > “The ablation studies are insufficient - while both dynamic weight scaling and attention stability Norm are introduced, their individual contributions to the final performance are not systematically isolated and analyzed.”
>
> **Response:**
> To quantitatively assess the contribution of each component (DWS and ASNorm), we systematically compared four variants. We have incorporated this discussion into **Sec 3.2 Figure 3**.
>
> 1. **Original Edge-Popup** (baseline)
> 2. **Edge-Popup + ASNorm**
> 3. **Edge-Popup + DWS**
> 4. **Edge-Popup + DWS + ASNorm** (our approach)
>
> We have presented these results in Figure 3 of the revised manuscript. Interestingly, while DWS alone decreases accuracy compared to the original Edge-Popup, the combination of DWS + ASNorm surpasses the baseline. This finding underscores the complementary nature of the two components—ASNorm stabilizes attention distributions, enabling DWS to balance weight magnitudes effectively.

---

> > ### Author Response · Authors · 2025-01-12
> > **Author Response for Reviewer odJi (Part 2)**
> >
> > **Q4. Add hyperparameter sensitivity analysis.**
> >
> > *Reviewer Comment:*
> > > “Add hyperparameter sensitivity analysis.”
> >
> > **Response:**
> > We agree that hyperparameter sensitivity is crucial. We provided a comprehensive plot in Figure 4 focusing on how varying sparsity levels affect performance. Following your suggestion, we have added new experiments to **Appendix D**, examining the impact of two additional parameters:
> >
> > - **Learning rate**
> > - **Number of layers**
> > - **Hidden dimensions**
> >
> > These experiments provide additional insights into how our approach performs under different configurations, further demonstrating the robustness of our method.
> >
> > ---
> >
> > **Q5. Add comprehensive hyperparameter settings.**
> >
> > *Reviewer Comment:*
> > > “Hyperparameter settings are missing.”
> >
> > **Response:**
> > Recognizing the importance of reproducibility, we have included a detailed table in **Appendix C (Table 6, 7, and 8)** listing all hyperparameters for each model and dataset, such as the number of layers, batch sizes, learning rates, and so forth. We hope these details facilitate easier reproduction of our results and foster further research on strong lottery tickets for Graph Transformers.
> >
> > ---
> >
> > **Q6. The OOD generalization claims, while promising, lack comparison to other robustness-enhancing techniques.**
> >
> > *Reviewer Comment:*
> > > “The OOD generalization claims, while promising, lack comparison to other robustness-enhancing techniques.”
> >
> > **Response:**
> > Most published works (e.g., Guo et al. [“Investigating Out-of-Distribution Generalization of GNNs: An Architecture Perspective”], Li et al. [“OOD-GNN: Out-of-Distribution Generalized Graph Neural Network”]) focus on **MPNN-based methods** such as GCN or GAT, typically relying on:
> >
> > 1. **Data augmentation or transformation** (e.g., graph Mixup, rewriting node features),
> > 2. **Novel architecture choices** (e.g., decoupled propagation, attention modules), or
> > 3. **Kernel-based regularizers** (e.g., Random Fourier Features).
> >
> > By contrast, our method does not require changing GT architectures or adding specialized modules; we propose an Adaptive Scaling that stabilizes random weights and attention distributions without modifying the fundamental Transformer blocks. Hence, our approach is complementary to existing OOD techniques.
> >
> > Moreover, because our technique simply scales the Transformer’s existing weights, it can be combined with other OOD strategies (e.g., data augmentations or invariant learning) without requiring an overhaul of the GT design. To the best of our knowledge, there are no existing approaches designed to enhance the OOD performance of Transformer-based GNNs. This flexibility distinguishes our work and addresses a gap in the literature, as no Transformer-specific OOD enhancements have yet been proposed.

---

### Review · Reviewer_hk8a · 2024-12-21

**Summary Of Contributions:**

This paper empirically studies the strong lottery ticket hypothesis in graph transformers.  This hypothesis states that there exists in a randomly initialized neural network at least one smaller subnetwork that has comparable performance to a trained full network.  This has been studied for other neural network architectures, but not for graph transformers.  The paper provides empirical evidence in favor of this hypothesis on various graph learning datasets.  To do this, the authors find that methods used for message passing neural networks are insufficient, yielding significant decreases in accuracy over dense, trained models.  They empirically trace this phenomenon to a tendency of previous methods to choose subnetworks with very sparse attention score matrices.  This leads them to design an adaptive scaling method, which mitigates this phenomenon.

They evaluate the produced lottery tickets in terms of accuracy on a variety of tasks in order to demonstrate their robust performance in comparison to dense, trained networks.

**Audience:**

Yes

**Claims And Evidence:**

Yes

**Requested Changes:**

1.) Can the authors provide intuition (of a somewhat theoretical variety) regarding the characteristics of a dataset that lead to existence of strong lottery tickets?  I feel that this discussion would strengthen the work.

**Strengths And Weaknesses:**

Strengths:

1.) The paper performs experiments to show the limitations of state of the art lottery ticket discovery methods.

2.) The paper probes the source of these limitations and uses it to devise a nontrivial modification, which demonstrates empirically improve performance.

3.) Various experiments show the benefits of strong lottery tickets in terms of improved robustness and model scalability.


Weaknesses:

1.) No theoretical results are included.  Thus, it is not clear what aspects of data-generating distributions lead one to expect the existence of strong lottery tickets or how large they are for a given level of accuracy.

---

> ### Author Response · Authors · 2025-01-12
> **Author Response for Reviewer hk8a**
>
> Thank you for your insightful feedback and valuable review.
>
> **Q1. Can the authors provide intuition (of a somewhat theoretical variety) regarding the characteristics of a dataset that lead to the existence of strong lottery tickets?**
>
> *Reviewer Comment:*
> > “It is not clear what aspects of data-generating distributions lead one to expect the existence of strong lottery tickets.”
>
> **Response:**
> Empirically, we have observed that the effectiveness of Strong Lottery Tickets (SLTs) varies substantially with dataset characteristics. It is possible to draw conclusions from Appendix B (Tables 3, 4, and 5). We have added this discussion to **Appendix B**:
>
> 1. **Task Type (Node vs. Graph Classification, Regression):**
>    - **Node Classification:** SLTs generally match or surpass dense models, as these tasks involve local regularities in node features and relationships. This local structure is more readily captured by subnetworks.
>    - **Graph Classification:** These tasks often require learning global structural features from diverse or irregular graphs, placing higher demands on representational capacity and reducing SLT effectiveness.
>    - **Regression (e.g., ZINC dataset):** A performance gap is observed between SLTs and dense models, likely due to the fine-grained outputs and structural complexity required.
>
> 2. **Graph Size and Complexity:**
>    - **Smaller, Simpler Graphs:** Datasets like MNIST-like graphs (fewer nodes/edges) show relatively smaller performance drops when using SLTs.
>    - **Larger, More Complex Graphs:** Datasets such as CIFAR10 or PATTERN exhibit larger gaps between SLTs and dense models. The increased node/edge count and intricate graph structures likely require denser connectivity to effectively capture global information.
>
> Overall, these findings suggest that SLTs are most effective when data distributions exhibit stronger local structure and lower global complexity. Tasks demanding fine-grained or globally integrated representations, as well as more complex graphs, tend to benefit from—or even require—larger models.
>
> ---
>
> **Q2. “It is not clear how large they are for a given level of accuracy.”**
>
> *Reviewer Comment:*
> > “It is not clear how large they are for a given level of accuracy.”
>
> **Response:**
> Our findings show that SLT performance depends on multiple factors—primarily layer depth, hidden dimension, and sparsity. We have added additional experiment results to **Appendix F (Table 12, 13)**:
>
> 1. **Sparsity Levels:**
>    - As illustrated in Figure 4, **moderate sparsity (10–90%)** yields SLT accuracy close to or even higher than the dense baseline.
>    - A steep drop is observed only at extreme sparsity levels (e.g., **99% sparsity**).
>
> 2. **Hidden Dimension:**
>    - From Table 13, SLTs remain competitive with dense models even at smaller hidden dimensions (e.g., **84.64% vs. 84.19%** at 8 dimensions).
>    - SLTs achieve their peak accuracy (**87.44%**) at 64 dimensions, surpassing the dense counterpart’s **86.85%**.
>
> 3. **Layer Depth:**
>    - Table 12 shows SLTs performing comparably to (or surpassing) dense networks with **2–3 layers.**
>    - Beyond 4 layers, SLT accuracy declines (e.g., **60.07% at 4 layers vs. 86.76% at 2 layers**).
>    - SLTs are more sensitive to the number of layers compared to dense models, suggesting the need to carefully select the appropriate number of layers.
>
> Overall, these experiments confirm that SLTs can remain remarkably compact (fewer layers, smaller hidden dimensions, and high sparsity) while preserving—or even improving—accuracy. The exact “size” required for a target accuracy thus depends on the interplay of architectural depth, dimensionality, and a suitably chosen sparsity regime.

---

### Review · Reviewer_fREY · 2024-12-29

**Summary Of Contributions:**

This paper introduces an approach to discover a small dense sub-network (lottery ticket) for graph transformers. The main contribution is not only the algorithm/approach, but also the confirmation of existing such a subnetwork.

**Audience:**

Yes

**Broader Impact Concerns:**

This can help address the sparsity issue of deep graph neural network design.

**Claims And Evidence:**

Yes

**Requested Changes:**

1. Adding discussions on the (strong) LT for general transformers; and its relationship to graph transformers
2. what are the theory of (strong) LT? Can they be applied to graph transformers?

**Strengths And Weaknesses:**

Strengths:
+ A simple yet effective approach for discovering the LT in graph transformers
+ The results confirm the existence of  the strong LT for graph transformers
+ The experimental results demonstrate the effectiveness of the proposed approach

Weaknesses:
- The paper in general lacks of technical depth
- I would like to see some analytical discussion of the strong LT for graph transformers
- The discussion of (strong) LT for general transformers should be added and why it cannot be utilized for graph transformers

---

> ### Author Response · Authors · 2025-01-12
> **Author Response for Reviewer fREY (Part 1)**
>
> Thank you for your insightful feedback and valuable review.
>
> **Q1. Address the Paper’s Technical Depth**
>
> *Reviewer Comment:*
> > “The paper in general lacks of technical depth.”
>
> **Response:**
> Building on feedback and to deepen the technical content, we now have strengthened the following points:
>
> 1. **Additional Pruning Comparisons (Appendix D, E):**
>    - We have compared our approach with multiple pruning baselines (N:M Sparsity, BitNet, and SLT methods such as UGTs and MF-GNN), clarifying the advantages and disadvantages of each technique from both empirical and algorithmic perspectives.
>
> 2. **Ablation Studies (Section 3.2, Fig. 3):**
>    - We have systematically analyzed the contributions of ASNorm and DWS in isolation and in combination. These ablations illustrate the interplay among different components, shedding light on why certain architectural designs can (or cannot) efficiently uncover LTs in Graph Transformers.
>
> 3. **Hyperparameter Sensitivity (Appendix F):**
>    - We have expanded the sensitivity analysis for additional hyperparameters (e.g., number of layers, learning rates, hidden dimensions), illustrating the parameter regimes under which Graph Transformer LTs are discovered effectively.
>
> These updates aim to address concerns regarding technical depth by providing a richer empirical and analytical understanding of the proposed approach.
>
> ---
>
> **Q2. Add discussions on the (strong) LT for general transformers and its relationship to graph transformers.**
>
> *Reviewer Comment:*
> > “The discussion of (strong) LT for general transformers should be added and why it cannot be utilized for graph transformers.”
>
> **Response:**
> Research has been conducted on topics such as **weak LT** [1] and **strong LT** [2] in general transformers. However, neither weak nor strong LT has been explored in the context of graph transformers. Therefore, this paper is the first to address strong LT in the field of graph transformers.
> This discussion and its implications have been added to **Section 2, Related Work**.
>
> General transformers and graph transformers share architectural components, such as attention mechanisms and feed-forward networks. However, the input data they handle—sequential text vs. graph structures—introduces differences in their behavior:
>
> - **Irregular Connectivity and Sparsity:** Graph transformers face unique challenges not present in the structured redundancies typical of NLP tasks.
> - **Failure of Naive Application:** While the SLT algorithm could be applied to graph transformers, its naive application, as done in [2] using methods like Edge-Popup, fails due to these challenges.
>
> In this work, we demonstrate that simply applying such methods is insufficient for graph transformers. To address these issues, we introduce **Adaptive Scaling**, a novel optimization approach tailored for graph transformers.
>
> References:
> [1] Brix et al., "Successfully Applying the Stabilized Lottery Ticket Hypothesis to the Transformer Architecture," *Proceedings of the 58th Annual Meeting of the Association for Computational Linguistics (ACL)*, 2020, pp. 3909–3915.
> [2] Shen et al., "What's Hidden in a One-layer Randomly Weighted Transformer?" *Proceedings of the 2021 Conference on Empirical Methods in Natural Language Processing (EMNLP)*, 2021, pp. 2914–2921.

---

> > ### Author Response · Authors · 2025-01-12
> > **Author Response for Reviewer fREY (Part 2)**
> >
> > **Q3. What are the theory of (strong) LT? Can they be applied to graph transformers? I would like to see some analytical discussion of the strong LT for graph transformers.**
> >
> > *Reviewer Comment:*
> > > “What are the theory of (strong) LT? Can they be applied to graph transformers?”
> >
> > **Response:**
> > We have summarized our current stance on the theory behind SLTs, how we position our work in this context, and why applying SLTs to graph transformers is both non-trivial and novel:
> >
> > 1. **Strong Lottery Tickets (SLT) Theory:**
> >    - As discussed in Section 2, SLT conjectures that accurate subnetworks exist within a randomly initialized network and can achieve comparable performance to the full network without any weight updates.
> >    - This concept has been studied in traditional neural architectures (e.g., fully connected and convolutional networks) where structured redundancies make such subnetworks easier to identify.
> >
> > 2. **Challenges in Applying SLTs to Graph Transformers:**
> >    - Graph Transformers process irregular graph structures with sparse connectivity, which differ fundamentally from the structured data handled by Transformers in NLP.
> >    - While the general SLTH could apply to Graph Transformers, our empirical findings show that existing methods like Edge-Popup fail in this context.
> >    - For instance, as shown in Figure 3 and Appendix B:
> >      - Learning global structural features from larger and more complex graphs (e.g., CLUSTER) sees significant degradation in accuracy by Edge-Popup compared to more regular graphs (e.g., OGBG-Molhiv).
> >      - Moreover, this precision gap becomes more pronounced as the number of layers increases.
> >        - On the ogbg-Molhiv dataset using a 10-layer model, the difference between Edge-Popup and Adaptive Scaling is **20%**.
> >        - On the CLUSTER dataset using a 16-layer model, the difference is **60%**.
> >
> > 3. **Novel Contribution of Adaptive Scaling:**
> >    - Our techniques successfully enable the discovery of SLTs in Graph Transformers by adapting to these irregularities, demonstrating competitive performance with dense models.
> >
> > Extending the existing theory of SLTs to Transformers, let alone Graph Transformers, is known to be non-trivial due to architectural differences and remains an open question. While this study focuses on empirical validation, theoretical exploration of SLTH for Graph Transformers remains a valuable direction for future work and is noted as a limitation in **Section 5**.

---

> ### Comment · Action_Editor_XABG · 2025-02-07
> **Please submit your official Recommendation asap**
>
> Dear Reviewer fREY,
>
> You are behind on this reviewing task. Could you please provide your official recommendation as soon as possible? The other two reviewers have already submitted their decisions.

---

> > ### Comment · Reviewer_hk8a · 2025-02-07
> >
> > Apologies.  I will submit this today.

---

### Author Response · Authors · 2025-01-12
**Summary of revision**

Thank you for your thoughtful feedback. In the manuscript, sentences that have been corrected or added are written in orange. Below is an overview of our main revisions:

---

### **1. Enhancing Technical Depth**

To address concerns regarding the technical depth of the paper:

- **Additional Baselines (Appendix D, E):**
  - We have added comparisons with pruning methods such as N:M Sparsity, BitNet, and SLT approaches.
  - These comparisons contextualize the advantages of our approach within the broader landscape of network compression techniques.

- **Expanded Ablation Studies (Section 3.2, Fig. 3):**
  - We have systematically isolated and analyzed the individual contributions of Dynamic Weight Scaling (DWS) and Attention Stability Norm (ASNorm).
  - These experiments demonstrate their complementary roles in uncovering Strong Lottery Tickets (SLTs) for Graph Transformers.

- **Hyperparameter Sensitivity Analysis (Appendix F):**
  - New experiments highlight the sensitivity of SLTs to factors such as sparsity levels, learning rates, number of layers, and hidden dimensions, providing practical guidance for SLT discovery.

---

### **2. Discussions on (Strong) Lottery Tickets (LTs) for General and Graph Transformers**

We have expanded **Section 2 (Related Work)** to include:

- A comprehensive discussion on the differences between general and graph transformers, highlighting why naive applications of SLT methods (e.g., Edge-Popup) fail for Graph Transformers.
- A theoretical summary of SLTs, detailing the challenges posed by irregular graph structures and how our proposed solutions (ASNorm and DWS) address them.

---

### **3. Dataset Characteristics and SLT Effectiveness**

We have investigated the influence of dataset characteristics on SLT performance (**Appendix B**):

- **Task Type:**
  - **Node classification tasks** show better SLT performance due to local regularities in node features and relationships.
  - **Graph classification tasks**, which require learning global structural features, remain more challenging for SLTs.

- **Graph Size and Complexity:**
  - SLTs perform better on smaller, simpler graphs and struggle with larger, more complex datasets that require dense connectivity for effective representation.

---

### **4. Reproducibility Enhancements**

To facilitate reproducibility:

- **Comprehensive hyperparameter settings** are now detailed in **Appendix C (Tables 6–8)**.
- **New experiments (Appendix F):**
  - These systematically analyze architectural and sparsity settings, providing a robust foundation for future research on SLTs in Graph Transformers.

---

We hope these revisions address your concerns and significantly strengthen the manuscript.

---

### Decision · Action_Editor_XABG · 2025-02-18

**Recommendation:** Accept as is

**Comment:**

This paper investigates the strong lottery ticket hypothesis (SLTH) in graph transformers, which posits that a randomly initialized network contains a smaller subnetwork capable of matching a trained full network's performance. While SLTH has been studied in other architectures, it remains unexplored for graph transformers. The authors provide empirical evidence supporting SLTH on graph learning datasets, revealing that methods for message-passing neural networks fail for graph transformers, often selecting subnetworks with overly sparse attention matrices and reducing accuracy. To address this, they propose an adaptive scaling method that mitigates the issue.

In the revision, the authors have addressed the reviewers' concerns by incorporating additional experiments to enhance technical depth, expanding discussions on the strong lottery ticket hypothesis (SLT), analyzing the influence of dataset characteristics, and providing more details to improve reproducibility. These revisions strengthen the paper's rigor and clarity, ensuring a more comprehensive and reliable contribution to the field.

**Audience:**

At least some individuals in TMLR's audience are interested in knowing the findings of this paper.

**Claims And Evidence:**

The claims made in the submission are supported by accurate, convincing and clear evidence.